# Rapid and definitive treatment of phenylketonuria in variant-humanized mice with corrective editing

Dominique L. Brooks[1,2,3], Manuel J. Carrasco [4], Ping Qu[1,2,3], William H. Peranteau[5,6], Rebecca C. Ahrens-Nicklas[7], Kiran Musunuru [1,2,3,9] ✉, Mohamad-Gabriel Alameh [4,8,9] ✉ & Xiao Wang [1,2,3,9] ✉

Phenylketonuria (PKU), an autosomal recessive disorder caused by pathogenic variants in the phenylalanine hydroxylase (*PAH*) gene, results in the accumulation of blood phenylalanine (Phe) to neurotoxic levels. Current dietary and medical treatments are chronic and reduce, rather than normalize, blood Phe levels. Among the most frequently occurring *PAH* variants in PKU patients is the P281L (c.842C>T) variant. Using a CRISPR prime-edited hepatocyte cell line and a humanized PKU mouse model, we demonstrate efficient in vitro and in vivo correction of the P281L variant with adenine base editing. With the delivery of ABE8.8 mRNA and either of two guide RNAs in vivo using lipid nanoparticles (LNPs) in humanized PKU mice, we observe complete and durable normalization of blood Phe levels within 48 h of treatment, resulting from corrective *PAH* editing in the liver. These studies nominate a drug candidate for further development as a definitive treatment for a subset of PKU patients.

Phenylalanine hydroxylase (PAH) deficiency prevents the conversion of Phe to tyrosine. Untreated PKU patients can have very high blood Phe levels of more than 1200 µmol/L (normal Phe levels are less than 120 µmol/L), with neurocognitive and neuropsychiatric sequelae. A strict low-Phe diet is the mainstay of treatment, with the goal of maintaining Phe levels of 120–360 µmol/L, which still exceed the physiologic range[1]. However, many PKU patients find it challenging to adhere to the unpalatable and cost-prohibitive diet. There are only two approved medical therapies. The first is sapropterin, an oral medication that serves as a cofactor of the PAH protein and can improve the activity of some mutant forms of PAH. However, the P281L variant, one of the most common PKU pathogenic variants with its highest prevalence in populations in the Middle East, Europe, and Russia (e.g.,

present in 14.8% of PKU patients in the Netherlands, 11.2% in Turkey, 10.8% in Portugal, 10.3% in Italy, and 9.4% in Germany)[2], is unresponsive to sapropterin (i.e., blood Phe levels do not improve with sapropterin treatment)[3]. The second is pegvaliase, an injectable enzyme that acts directly to catabolize Phe. However, pegvaliase carries a substantial risk of anaphylaxis and has a black box warning from the U.S. Food and Drug Administration on its label for that reason. In addition, the dosing regimen is complex, starting with once-weekly injections and slowly up-titrating to once-daily injections, and it is available in the U.S. only through a Risk Evaluation and Mitigation Strategy program. In a long-term study, patients on pegvaliase achieved a mean 51% Phe reduction at 1 year after initiation (1233–565 µmol/L)[4]. Thus, a safe one-time therapy that would durably,

[1]Cardiovascular Institute, Perelman School of Medicine at the University of Pennsylvania, Philadelphia, Pennsylvania, USA. [2]Division of Cardiovascular Medicine, Department of Medicine, Perelman School of Medicine at the University of Pennsylvania, Philadelphia, Pennsylvania, USA. [3]Department of Genetics, Perelman School of Medicine at the University of Pennsylvania, Philadelphia, Pennsylvania, USA. [4]Department of Bioengineering, George Mason University, Fairfax, Virginia, USA. [5]The Center for Fetal Research, Children's Hospital of Philadelphia, Philadelphia, Pennsylvania, USA. [6]Division of Pediatric General, Thoracic, and Fetal Surgery, Children's Hospital of Philadelphia, Philadelphia, Pennsylvania, USA. [7]Division of Human Genetics and Metabolism, Children's Hospital of Philadelphia, Philadelphia, Pennsylvania, USA. [8]Division of Infectious Diseases, Department of Medicine, Perelman School of Medicine at the University of Pennsylvania, Philadelphia, Pennsylvania, USA. [9]These authors jointly supervised this work: Kiran Musunuru, Mohamad-Gabriel Alameh, Xiao Wang. ✉e-mail: kiranmusunuru@gmail.com; Mg.Alameh@pennmedicine.upenn.edu; xiao8@pennmedicine.upenn.edu

even permanently, normalize blood Phe levels would be a vastly superior treatment option over existing alternatives. Although the liver is spared from toxicity in PKU, the *PAH* gene is largely expressed in hepatocytes, and correction of the primary genetic defect solely within the liver would in principle be curative in PKU patients. Previous studies suggested that restoration of ≈10% normal PAH function would suffice[5–9].

In vivo CRISPR editing is an emerging new therapeutic approach to directly correct pathogenic variants in a patient's own body in organs such as the liver. Base editors[10,11] and prime editors[12] are advantageous for the correction of variants because they can function efficiently and precisely without the need for double-strand breaks. Although in vivo base-editing therapeutics have been demonstrated to be effective in non-human primates[13–15] and are now being tested in human patients[16], these drugs are intended to introduce variants into wild-type genes, allowing for the use of wild-type models for pre-clinical studies. A challenge in developing corrective base-editing and prime-editing therapeutics is the lack of readily available in vitro and in vivo models harboring rare patient-specific variants, needed for establishing the efficacy and safety of drug candidates.

In this work, we executed a workflow to generate in vitro and in vivo models in order to develop and validate a corrective therapeutic for the *PAH* P281L variant. After screening candidate adenine base editors (ABEs) and guide RNAs (gRNAs) for corrective editing efficacy and off-target editing using an in vitro P281L cellular model, we demonstrated rapid and definitive treatment of PKU in homozygous and compound heterozygous humanized P281L mouse models with two ABE/gRNA combinations, delivered via lipid nanoparticles (LNPs) into the liver.

## Results

### In vitro studies

We focused on the P281L (c.842C>T) variant in *PAH* exon 7 because of features that suggested it would be amenable to therapeutic adenine base editing (Fig. 1a): correction with an A-to-G change on the antisense strand; two candidate gRNAs (designated PAH1 and PAH2) with NGG protospacer-adjacent motifs (PAMs) compatible with *Streptococcus pyogenes* Cas9 (SpCas9) ABEs; positioning of the target adenine in the editing windows of SpCas9 base editors (protospacer position 5 with PAH1, position 4 with PAH2); and only a single additional adenine base that might have undesired bystander editing (protospacer position 3 with PAH1, position 2 with PAH2). We used prime editing, specifically PE5max with an engineered pegRNA[17,18], to introduce the P281L variant into HuH-7 human hepatoma cells, a commonly used proxy for primary human hepatocytes. Observing ≈30% editing in bulk HuH-7 cells, we used single-cell cloning to identify and expand a P281L homozygous HuH-7 cell line (Fig. 1b). We used this cell line to screen for the on-target activity of a variety of ABEs[19,20] in combination with either PAH1 or PAH2 in plasmid transfection experiments (Fig. 1c). Although all ABE/gRNA sets produced a substantial level of editing, the most favorable combinations of maximal on-target corrective editing and minimal bystander editing were produced by ABE8.8 with the two gRNAs, in each case achieving corrective editing without bystander editing on ≈50% of the alleles. ABE8.8/PAH1 had greater though still low-level bystander editing compared to ABE8.8/PAH2. All other ABE/gRNA sets, particularly those with ABE8e, had substantially greater bystander editing, reflecting their wider editing windows. A validated, highly efficient, positive control ABE8.8/gRNA set (targeting the *PCSK9* gene[13,14]) produced ≈60% on-target editing in these cells.

We generated ABE8.8 mRNA with in vitro transcription and formulated LNPs with ABE8.8 mRNA and either of two chemically synthesized oligonucleotides, the PAH1 gRNA or the PAH2 gRNA. We performed dose-response studies with the LNPs using the P281L homozygous HuH-7 cell line (Fig. 1d); there was essentially 100% corrective editing at higher doses, with virtually identical EC$_{50}$ values for

the two gRNAs, establishing equivalent potency in vitro. As anticipated, PAH1 showed low levels of bystander editing at higher doses. We noted that the P281L variant is located at the junction of exon 7 and intron 7 and thus might disrupt mRNA splicing. With quantitative RT-PCR using primers in *PAH* exon 7 and exon 8, with primers in *PAH* exon 1 and exon 2 serving as reference, we observed five- to sixfold reduced levels of exon 7/exon 8 transcripts in P281L homozygous HuH-7 cells compared to wild-type cells (Fig. 1e). Treatment with either LNP formulation (≈100% corrective editing) fully rescued mRNA splicing, whereas plasmid transfection (≈50% corrective editing) resulted in partial rescue (Fig. 1e).

### Off-target assessment

To evaluate ABE8.8/PAH1 and ABE8.8/PAH2 off-target editing, we performed ONE-seq[13,21] with synthetic human genomic libraries selected by homology to the PAH1 or PAH2 protospacer sequence, treated with recombinant ABE8.8 protein and the appropriate gRNA, and assessed the top 48 ONE-seq-nominated sites with next-generation sequencing of targeted PCR amplicons from ABE8.8/gRNA plasmid-transfected versus control-transfected P281L homozygous HuH-7 genomic DNA samples (Fig. 1f; Supplementary Tables 1 and 2). With ABE8.8/PAH2, we observed a site with low-level off-target base editing (≈0.2%) and a site with substantial off-target base editing (≈8%). In contrast, among the top sites for ABE8.8/PAH1, we observed only one site (PAH1_OT7) with borderline off-target editing (≈0.1%). Using P281L homozygous HuH-7 cells treated with a very high dose of ABE8.8/PAH1 LNPs (2500 fg/cell), we assessed the same top sites (Supplementary Table 3) and again observed only one site (PAH1_OT7) with borderline off-target editing (≈0.1%). This one site lies in an intron in the *SPAG17* gene, which encodes a ciliary structural protein and thus is not concerning for oncogenic risk. When we assessed for off-target editing at PAH1_OT7 in the ABE8.8/PAH1 LNP dose-response HuH-7 samples, we observed PAH1_OT7 editing (≈0.1%) only at the highest dose, a far higher exposure than would be experienced by hepatocytes in vivo (Fig. 1g). Accordingly, we favor ABE8.8/PAH1 as a drug candidate and have prioritized in vivo studies with ABE8.8/PAH1.

### In vivo studies

For in vivo studies, we used CRISPR-Cas9 targeting in mouse embryos to generate a humanized PKU model, in the C57BL/6J background, in which we replaced a small portion of the endogenous mouse *Pah* exon 7 with the orthologous human sequence spanning the PAH1 and PAH2 protospacer/PAM sequences and containing the P281L variant (Fig. 2a). Homozygous P281L mice had phenotypes consistent with PKU, including hypopigmentation (reduced melanin synthesis due to decreased tyrosine levels) (Fig. 2b). In a short-term study, we treated four age-matched (8 weeks of age) homozygous P281L (PKU) mice with ABE8.8/PAH1 LNPs, with three heterozygous P281L (non-PKU) colonymates and three untreated homozygous P281L (PKU) colonymates serving as controls (Fig. 3a). At baseline, the PKU mice had blood Phe levels ranging from 1455 to 2242 μmol/L, whereas the non-PKU mice had blood Phe levels <120 μmol/L (similar to human profiles). Some of the treated mice displayed substantially decreased Phe levels at 24 h after treatment (36% mean reduction for all treated mice). All the treated mice had largely normalized Phe levels at 48 h after treatment (90% mean reduction) and were indistinguishable from non-PKU mice at 1 week after treatment (PKU, mean 104 μmol/L; non-PKU, mean 96 μmol/L).

We performed two additional short-term studies. In one study, we treated two 4-week-old compound heterozygous P281L (PKU) mice (with a non-humanized second mutant allele, ΔGTAA, generated by non-homologous end-joining at the P281L target site) with ABE8.8/PAH1 LNPs, with two heterozygous P281L (non-PKU) littermates serving as controls (Fig. 3b). In the other study, we treated two 8-week-old homozygous P281L (PKU) mice and three 8-week-old compound

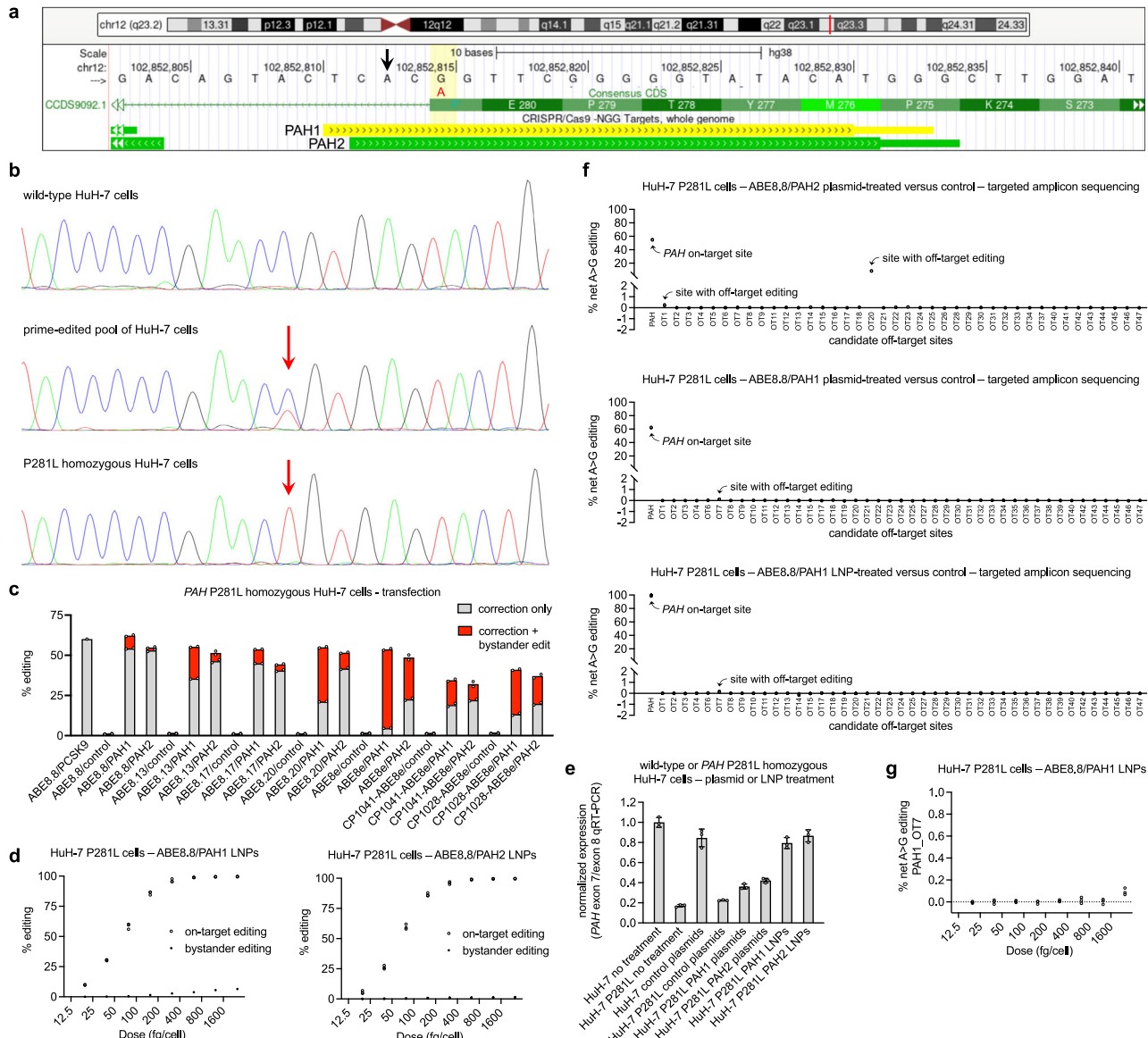

**Fig. 1 | Adenine base editing for correction of *PAH* P281L variant in human hepatocytes in vitro. a** Schematic of genomic site of the *PAH* P281L variant, adapted from UCSC Genome Browser (GRCh38/hg38). The vertical yellow bar indicates the G altered to A (in red) by the variant. The A two positions upstream of the variant (black arrow) is a potential site of bystander editing. The horizontal yellow bar and green bar indicate protospacer (thick) and PAM (thin) sequences targeted by PAH1 gRNA and PAH2 gRNA, respectively. **b** Generation of *PAH* P281L homozygous HuH-7 cell line. Top, sequence from wild-type HuH-7 cells. Middle, sequence from a pool of HuH-7 cells following prime editing. Bottom, sequence from an isolated clonal line homozygous for the P281L variant (HuH-7 P281L cells). Red arrows indicate the P281L site. **c** A-to-G editing following transfection of HuH-7 P281L cells with plasmids encoding ABE/gRNA combinations (*n* = 2 biological replicates). **d** A-to-G editing observed in dose-response studies with HuH-7 P281L cells treated with ABE8.8/PAH1 LNPs (left) or ABE8.8/PAH2 LNPs (right). On-target

editing includes all outcomes with corrective P281L editing, irrespective of bystander editing (*n* = 3 biological replicates). **e** qRT-PCR from HuH-7 samples using flanking primers in *PAH* exons 7 and 8 and flanking primers in *PAH* exons 1 and 2, displaying the quantified ratio of exon 7/exon 8 to exon 1/exon 2 (*n* = 3 biological replicates; mean ± standard deviation for each condition, normalized to untreated wild-type HuH-7 cells). **f** On-target or off-target editing at top ONE-seq-nominated candidate sites calculated as net A-to-G editing (proportion of sequencing reads with alteration of ≥1 A bases to G in treated samples versus untreated samples) in HuH-7 P281L cells that underwent plasmid transfection (top two graphs; *n* = 2 treated and 2 untreated biological replicates) or LNP treatment (bottom graph; *n* = 3 treated and 3 untreated biological replicates). Sites with unsuccessful sequencing are omitted. **g** Off-target editing at PAH1_OT7 site in a dose-response study with HuH-7 P281L cells treated with ABE8.8/PAH1 LNPs (*n* = 3 biological replicates). Source data are provided as a Source Data file.

heterozygous P281L (PKU) mice with ABE8.8/PAH2 LNPs, with three heterozygous P281L (non-PKU) littermates serving as controls (Fig. 3c). In all cases, the treated PKU mice had largely normalized blood Phe levels by 48 h after treatment (91 and 86% mean reductions for second and third short-term studies, respectively).

We have maintained three ABE8.8/PAH1 LNP-treated PKU mice and three control non-PKU mice from the first short-term study in an ongoing long-term study; up to 24 weeks after treatment, the LNP-

treated mice have maintained normal Phe levels (Fig. 3d). By 8 weeks after treatment, the hypopigmentation of the treated PKU mice had resolved (Fig. 2b). Aspartate aminotransferase (AST) and alanine aminotransferase (ALT) levels had slight rises in some mice at 24 h after treatment, remaining within the normal ranges and resolving by 72 h (Fig. 4a). (Transient AST and ALT rises are often observed with LNP treatments[13], although the identities of the lipid and/or RNA components responsible for the rises remain unresolved.)

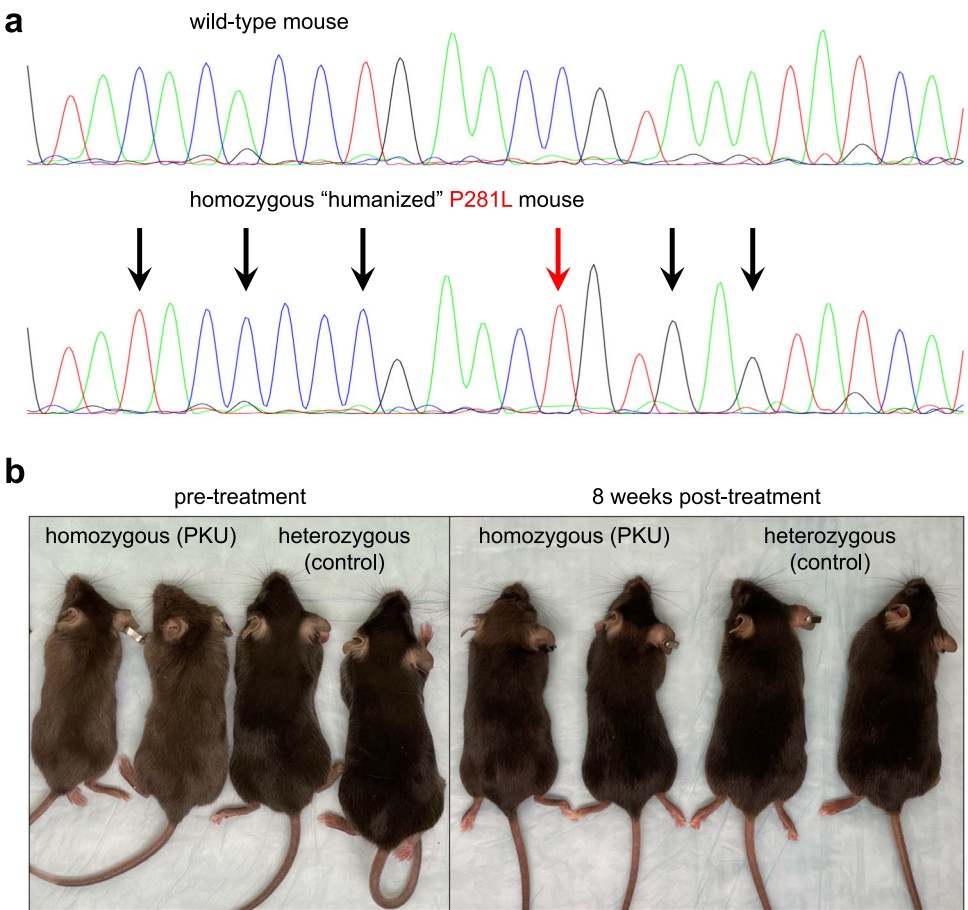

**Fig. 2 | Generation of humanized PKU mice with *PAH* P281L variant. a** Sanger sequencing chromatograms showing the generation of a humanized mouse model via Cas9-mediated homology-directed repair in mouse zygotes. At the top is the sequence from a wild-type C57BL/6J mouse. At the bottom is the sequence from a mouse homozygous for the humanized *Pah* P281L allele. The red arrow indicates the site of the P281L variant, and the black arrows indicate the sites of synonymous changes that humanize the local region of the mouse *Pah* gene. **b** Age-matched colonymates that are homozygous or heterozygous for the humanized P281L allele. The left picture shows two homozygous mice with PKU as evidenced by hypopigmentation of the fur and two control heterozygous mice with normal fur color, immediately prior to treatment. The right picture shows the two homozygous mice and two heterozygous mice 8 weeks after the homozygous mice received LNP treatment, with normalization of fur color.

We necropsied four ABE8.8/PAH1 LNP-treated homozygous mice (including one from the original short-term study), two ABE8.8/PAH1 LNP-treated compound heterozygous mice (second short-term study), and five ABE8.8/PAH2 LNP-treated mice (third short-term study) 1–2 weeks after treatment to assess editing in the liver and a variety of other organs. We observed that corrective editing occurred predominantly in the liver, with low-level editing observed in the spleen and minimal editing in the other organs, consistent with prior LNP studies (Fig. 3e). The desired corrective editing in the liver with ABE8.8/PAH1 LNPs ranged from 28 to 46% in the homozygous mice and from 26 to 52% of the editable alleles in the compound heterozygous mouse (i.e., 13 to 26% of total alleles); with ABE8.8/PAH2 LNPs, 47 to 58% of the editable alleles (Fig. 3e). Very low levels of bystander editing were observed (mean 0.8% with PAH1, mean 0.6% with PAH2). Liver histology from the necropsied ABE8.8/PAH1 LNP-treated mouse from the first short-term study showed no evidence of pathology (Fig. 4b).

## Discussion

In prior studies of CRISPR editing to correct the non-human variant present in the *Pah^enu2* PKU mouse model, using AAV or adenoviral vectors for delivery, mice did not experience normalization of blood Phe levels until 1–2 months after treatment (prolonged cytosine base editing[8]) or failed to achieve normalization (prime editing[9]). In this study, we achieved corrective editing of a frequent human PKU pathogenic variant and the complete normalization of blood Phe levels in mice within a few days of LNP treatment. We note the advantages of LNP editing therapeutics over viral vector editing therapeutics, including the lack of prolonged expression of the editor—which can elicit cytolytic immune responses and exacerbate off-target editing— and lack of risk of vector sequence integration, as well as the ability to redose the therapy if needed. Indeed, a study in which *Pah^enu2* PKU mice were treated with LNPs to deliver a cytosine base editor into the liver did not achieve normalization of blood Phe levels with a single LNP dose, but redosing with the same LNP treatment a week later resulted in increased cumulative editing and achieved near-normalization of blood Phe levels[22]. We further note that LNP-mediated CRISPR editing has achieved success in the clinic[23] and that an LNP drug product with ABE8.8 mRNA and a gRNA targeting *PCSK9* has demonstrated highly efficacious and durable hepatic editing in non-human primates[13] and has been safely dosed in patients in a clinical trial[16]. Based on the results reported here, in principle, the same exact LNP drug product with only the first 20 nucleotides of the gRNA component switched from the *PCSK9* sequence to the PAH1 sequence could achieve rapid, durable normalization of blood Phe levels in PKU patients with at least one copy of the P281L variant.

Thus, we are poised to initiate the development of an ABE8.8/ PAH1 LNP therapy for clinical use. A "one-and-done" Phe-normalizing

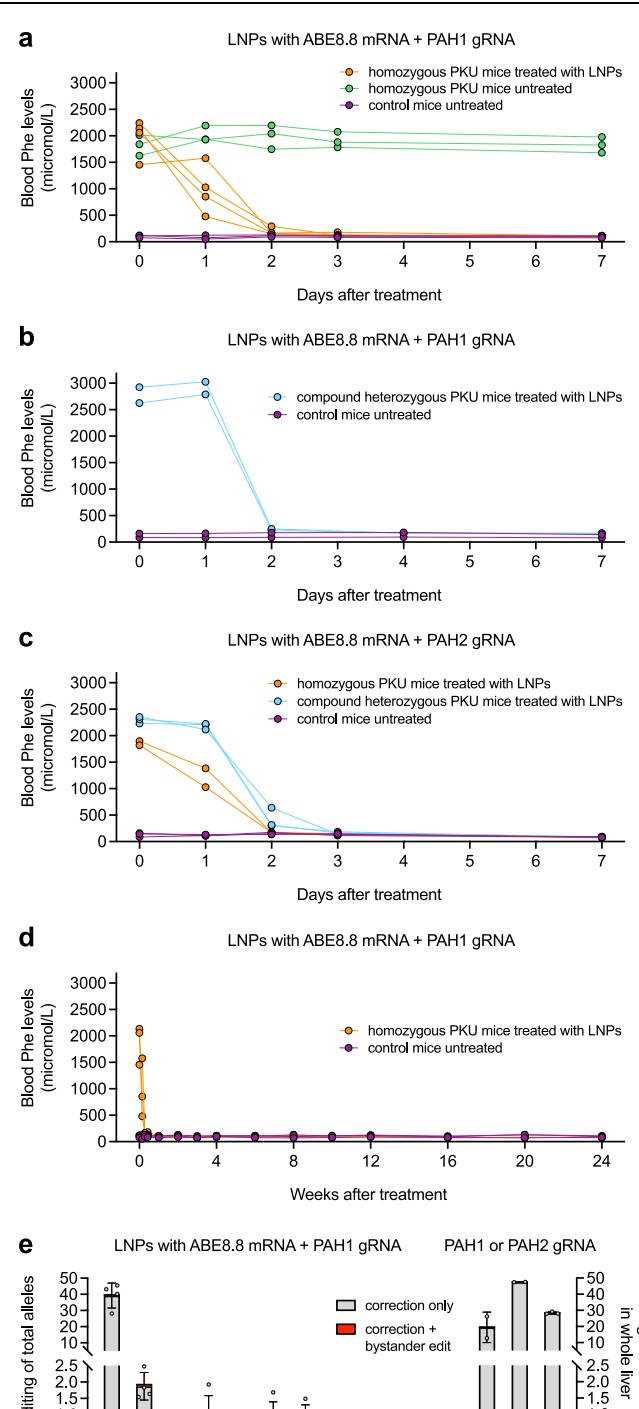

**Fig. 3 | Adenine base editing for correction of *PAH* P281L variant in humanized PKU mice in vivo. a** Short-term changes in the blood phenylalanine level in homozygous PKU mice (*n* = 4 animals) following treatment with 2.5 mg/kg dose of ABE8.8/PAH1 LNPs, comparing levels at various time points up to 7 days following treatment to levels in untreated PKU (*n* = 3 animals) and heterozygous non-PKU control (*n* = 3 animals) age-matched (8 weeks of age) colonymates (1 blood sample per timepoint). **b** Short-term changes in the blood phenylalanine level in compound heterozygous PKU mice (*n* = 2 animals) following treatment with 2.5 mg/kg dose of ABE8.8/PAH1 LNPs, comparing levels at various time points up to 7 days following treatment to levels in untreated heterozygous non-PKU control (*n* = 2 animals) age-matched (4 weeks of age) colonymates (1 blood sample per timepoint). **c** Short-term changes in the blood phenylalanine level in homozygous PKU mice (*n* = 2 animals) and compound heterozygous PKU mice (*n* = 3 animals) following treatment with 2.5 mg/kg dose of ABE8.8/PAH2 LNPs, comparing levels at various time points up to 7 days following treatment to levels in untreated heterozygous non-PKU control (*n* = 3 animals) age-matched (8 weeks of age) colonymates (1 blood sample per timepoint). **d** Long-term changes in the blood phenylalanine level in homozygous PKU mice (*n* = 3 animals) following treatment with 2.5 mg/kg dose of ABE8.8/PAH1 LNPs, comparing levels at various time points up to 24 weeks following treatment to levels in untreated heterozygous non-PKU control (*n* = 3 animals) age-matched (8 weeks of age) colonymates (1 blood sample per timepoint). **e** A-to-G editing in various mouse organs (left, *n* = 4 animals except for testis, for which *n* = 2 male animals; right, *n* = 2 or 3 animals per group), assessed 1–2 weeks following treatment with 2.5 mg/kg dose of LNPs (mean ± standard deviation for each organ). For compound heterozygous mice, each displayed number is % edited P281L alleles (editable alleles) divided by two. Source data are provided as a Source Data file.

Biosafety Committee at the University of Pennsylvania, where the studies were performed, and were consistent with local, state, and federal regulations as applicable, including the National Institutes of Health Guidelines for Research Involving Recombinant or Synthetic Nucleic Acid Molecules. All procedures used in animal studies were approved by the Institutional Animal Care and Use Committee at the University of Pennsylvania (protocol #805887), where the studies were performed, and were consistent with local, state, and federal regulations as applicable, including the National Institutes of Health Guide for the Care and Use of Laboratory Animals.

## Plasmids
The following ABE-expressing plasmids were obtained from Addgene as gifts from Dr. Nicole Gaudelli[19]: ABE8.8 (ABE8.8-m; Addgene plasmid # 136294; RRID:Addgene_136294), ABE8.13 (ABE8.13-m, Addgene plasmid # 136296; RRID:Addgene_136296), ABE8.17 (ABE8.17-m; Addgene plasmid # 136298; RRID:Addgene_136298), and ABE8.20 (ABE8.20-m; Addgene plasmid # 136300; RRID:Addgene_136300). The following ABE-expressing and prime editor-expressing plasmids were obtained from Addgene as gifts from Dr. David Liu[17,18,20]: ABE8e (ABE8e; Addgene plasmid # 138489; RRID:Addgene_138489), CP1041-ABE8e (CP1041-ABE8e; Addgene plasmid # 138493; RRID:Addgene_138493), CP1028-ABE8e (CP1028-ABE8e; Addgene plasmid # 138492; RRID:Addgene_138492), PEmax (pCMV-PEmax-P2A-hMLH1dn; Addgene plasmid # 174828; RRID:Addgene_174828), and pU6-tevopreq1-GG-acceptor (pU6-tevopreq1-GG-acceptor; Addgene plasmid # 174038; RRID:Addgene_174038); an epegRNA was expressed from the last plasmid, with the following spacer sequence (plus an additional 5′ G to facilitate U6 expression) and PBS/RTT sequence cloned into the plasmid: 5′-GTAGCTGGAGGACAGTACTCA-3′ and 5′-ACCCCCGAACTGT-GAGTACTGTCCTCCA-3′. gRNAs were expressed from the pGuide plasmid (pGuide; Addgene plasmid # 64711; RRID:Addgene_64711), with each of the following spacer sequences (plus an additional 5′ G to facilitate U6 expression) cloned into the plasmid: PAH1, 5′-GTCA-CAGTTCGGGGGTATACA-3′; PAH2, 5′-GCACAGTTCGGGGGTATACAT-3′; PCSK9, 5′-GCCCGCACCTTGGCGCAGCGG -3′; and nicking gRNA for prime editing, 5′-GTCCTCCAGCTACCAGTTGCC-3′.

therapy would offer substantial advantages over current chronic, lifetime, Phe-reducing treatments, and for the relevant patients, could represent a new standard of care. We anticipate that similar base-editing or prime-editing therapies could be developed for other PKU pathogenic variants, as well as a variety of other inborn errors of metabolism.

## Methods
The research described here complied with all relevant regulations. All recombinant DNA research was approved by the Institutional

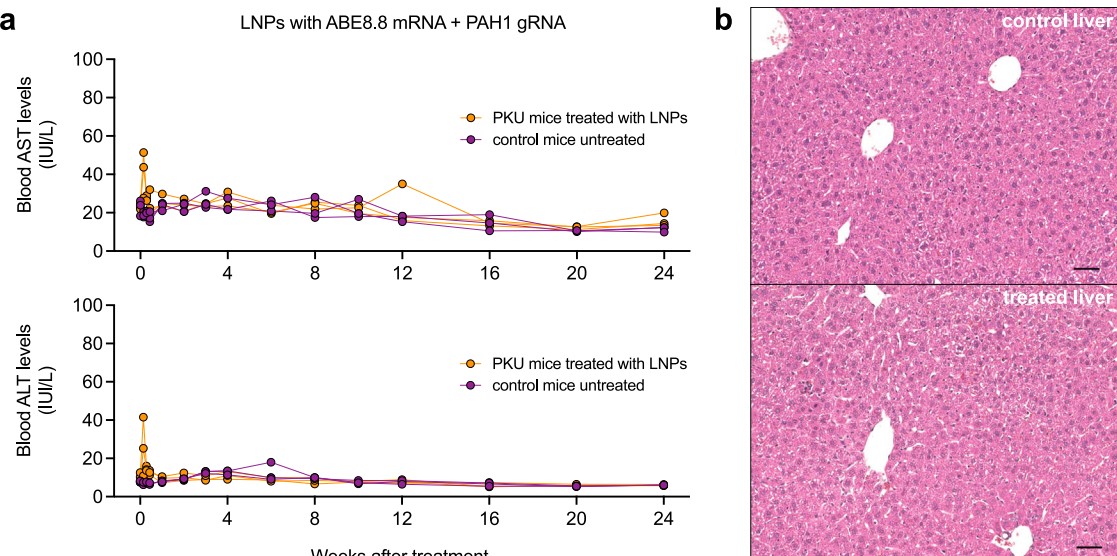

**Fig. 4 | Assessment of mouse liver following LNP treatment. a** Long-term changes in the blood aspartate aminotransferase (AST) level (top) and alanine aminotransferase (ALT) level (bottom) in homozygous PKU mice (*n* = 3 animals) following treatment with 2.5 mg/kg dose of ABE8.8/PAH1 LNPs, comparing levels at various time points up to 24 weeks following treatment to levels in untreated heterozygous non-PKU control (*n* = 3 animals) age-matched (8 weeks of age) colonymates (1 blood sample per timepoint). **b** Liver histology (hematoxylin/eosin staining) at ×20 magnification upon necropsy at 1 week after LNP treatment of humanized PKU mouse (bottom, *n* = 1 animal) compared to age-matched, untreated non-PKU mouse (top, *n* = 1 animal). Lines indicate a distance of 50 μm. Source data are provided as a Source Data file.

## RNA production

100-mer PAH1 and PAH2 gRNAs were chemically synthesized under solid phase synthesis conditions by a commercial supplier (Agilent) with end-modifications as well as heavy 2′-O-methylribosugar modification as previously described[24]: PAH1, 5′-mU*mC*mA*CAGUUCGG GGGUAUACAGUUUUAGAmGmCmUmAmGmAmAmAmUmAmGmCA AGUUAAAAUAAGGCUAGUCCGUUAUCAmAmCmUmUmGmAmAm AmAmAmGmUmGmGmCmAmCmCmGmAmGmUmCmGmGmUmG mCmU*mU*mU*mU-3′; PAH2, 5′-mC*mA*mC*AGUUCGGGGGUAUACA UGUUUUAGAmGmCmUmAmGmAmAmAmUmAmGmCAAGUUAAAA UAAGGCUAGUCCGUUAUCAmAmCmUmUmGmAmAmAmAmAmGm UmGmGmCmAmCmCmCmGmAmGmUmCmGmGmUmGmCmU*mU*-mU*mU-3′; where "m" and * respectively indicate 2′-O-methylation and phosphorothioate linkage. ABE8.8 mRNA was produced via in vitro transcription (IVT) and purification. In brief, a plasmid DNA template containing a codon-optimized ABE8.8 coding sequence and a 3′ poly-adenylate sequence was linearized. An IVT reaction containing linearized DNA template, T7 RNA polymerase, NTPs, and cap analog was performed to produce mRNA containing N1-methylpseudouridine. After digestion of the DNA template with DNase I, the mRNA product underwent purification and buffer exchange, and the purity of the final mRNA product was assessed with spectrophotometry and capillary gel electrophoresis. The elimination of double-stranded RNA contaminants was assessed using dot blots and transfection into human dendritic cells. Endotoxin content was measured using a chromogenic Limulus amebocyte lysate (LAL) assay; all assays were negative.

## LNP formulation

LNPs were formulated as previously described[25], with the lipid components (SM-102, 1,2-distearoyl-sn-glycero-3-phosphocholine, cholesterol, and PEG-2000 at molar ratios of 50:10:38.5:1.5) being rapidly mixed with an aqueous buffer solution containing ABE8.8 mRNA and either PAH1 gRNA or PAH2 gRNA in a 1:1 ratio by weight in 25 mM sodium acetate (pH 4.0), with an N:P ratio of 5.6. The resulting LNP formulations were subsequently dialyzed against sucrose-containing buffer, concentrated using Amicon Ultra-15 mL Centrifugal Filter Units (Millipore Sigma), sterile-filtered using 0.2-μm filters, and frozen until use. The LNPs had particle sizes of 69–89 nm (Z-Ave, hydrodynamic diameter), with a polydispersity index of <0.21 as determined by dynamic light scattering (Malvern NanoZS Zetasizer) and 90–100% total RNA encapsulation as measured by the Quant-iT Ribogreen Assay (Thermo Fisher Scientific).

## Culture and transfection of HuH-7 cells

HuH-7 cells were maintained in Dulbecco's modified Eagle's medium (containing 4 mM L-glutamine and 1 g/L glucose) with 10% fetal bovine serum and 1% penicillin/streptomycin at 37 °C with 5% CO$_2$. HuH-7 cells were seeded on 6-well plates (Corning) at $3.5 \times 10^5$ cells per well. At 16–24 h after seeding, cells were transfected at approximately 80–90% confluency with 9 μL TransIT®-LT1 Transfection Reagent (MIR2300, Mirus), 2 μg base editor plasmid, and 1 μg gRNA plasmid per well according to the manufacturer's instructions; alternatively, LNPs were added at various doses (quantified by the total amount of RNA within the LNPs) directly to the media. Cells were cultured for 72 h after transfection, and then media were removed, cells were washed with 1× DPBS (Corning), and genomic DNA was isolated using the DNeasy Blood and Tissue Kit (QIAGEN) according to the manufacturer's instructions.

## Generation of a *PAH* P281L homozygous HuH-7 cell line with prime editing

HuH-7 cells in a well of a 6-well plate were transfected with 9 μL TransIT®-LT1 Transfection Reagent, 1.5 μg PEmax plasmid, 0.75 μg epegRNA-expressing plasmid, and 0.75 μg nicking gRNA plasmid. Cells were dissociated with trypsin 48 h post-transfection and replated onto 10-cm plates (5000 cells/plate) with conditioned medium to facilitate recovery, and genomic DNA was isolated from the remainder of the cells as a pool to perform PCR and Sanger sequencing of the *PAH* P281L site. Single cells were permitted to expand for 7–14 days to establish clonal populations. Colonies were manually picked and replated into individual wells of a 96-well plate. Genomic DNA was isolated from individual clones and PCR and Sanger sequencing was performed to identify P281L homozygous HuH-7 clones. One representative clone was expanded for use in

subsequent studies. This P281L homozygous HuH-7 cell line is readily available from the authors via a Transfer of Research Material agreement with the University of Pennsylvania.

## Generation of a humanized PKU mouse model

The PKU mouse model with one or more humanized *Pah* P281L alleles was generated using in vitro transcribed Cas9 mRNA, a synthetic gRNA (spacer sequence 5′-UAGCUGAAGAAUGAUACUUA-3′) (Integrated DNA Technologies), and a synthetic single-strand DNA oligonucleotide (Integrated DNA Technologies) with homology arms matching the target site and harboring the P281L variant and synonymous variants (bold with underline): 5′-TGCTGGCTTACTGTCGTCTCGAGATTTCTT GGGTGGCCTGGCCTTCCGAGTCTTCCACTGCACACAGTACATTAGGC ATGGATCTAAGCCCATGTA**T**AC**CCCC**GAAC**T**GT**GAG**TATCATTCTTCA GCTACCCCTGCCAACCACAATGGATGCTCAAAGAATGCTGATCAGGC TCATTGCAGGCTGGTCCCCATGATCCAC-3′. The mixture of the three components was injected into the cytoplasm of fertilized oocytes from C57BL/6J mice at the Penn Vet Transgenic Mouse Core (https://www. vet.upenn.edu/research/core-resources-facilities/transgenic-mouse-core). Genomic DNA samples from founders were screened for knock-in of the desired sequence in the *Pah* locus via homology-directed repair. Founders with the humanized P281L allele were bred through two generations to obtain homozygous mice. In some founders, indel mutations were present because of non-homologous end-joining within the mouse *Pah* locus; a non-humanized loss-of-function allele with a 4-bp deletion (ΔGTAA) just distal to the site of the P281L variant was bred together with the humanized P281L allele to generate compound heterozygote mice. This humanized PKU mouse model is readily available from the authors via a Transfer of Research Material agreement with the University of Pennsylvania.

## Mouse studies

Mice were maintained on a 12-h light/12-h dark cycle, with a temperature range of 65–75 °F and a humidity range of 40–60%, and were fed ad libitum with a chow diet (LabDiet, Laboratory Autoclavable Rodent Diet 5010). Homozygous and compound heterozygous humanized PKU mice, as well as heterozygous humanized non-PKU mice, were generated as littermates/colonymates via timed breeding, in some cases using wild-type C57BL/6J mice (stock no. 000664) obtained from The Jackson Laboratory. Genotyping was performed using PCR amplification from genomic DNA samples (prepared from clipped tails/ears) followed by next-generation sequencing. Age-matched female and male colonymates were used for experiments at 4 weeks of age or 8 weeks of age with random assignment of animals to various experimental groups when applicable and with collection and analysis of data performed in a blinded fashion when possible. LNPs were administered to the mice at approximately 2.5 mg/kg doses via retro-orbital injection under anesthesia with 1–2% inhaled isoflurane. In short-term studies, mice were euthanized at 1–2 weeks after treatment, and eight liver samples (two from each lobe) and samples of other organs were obtained on necropsy and processed with the DNeasy Blood and Tissue Kit (QIAGEN) as per the manufacturer's instructions to isolate genomic DNA. Euthanasia in all instances was achieved via terminal inhalation of carbon dioxide followed by secondary euthanasia through cervical dislocation or decapitation, consistent with the 2020 American Veterinary Medical Association Guidelines on Euthanasia. Next-generation sequencing results from the liver samples were averaged to provide quantification of whole-liver editing. In both short-term and long-term studies, blood samples were collected via the tail tip at various time points (pre-treatment, day 1, day 2, day 3 or 4, day 7, and—when applicable—day 14, day 21, day 28, week 6, week 8, week 10, week 12, week 16, week 20, and week 24), in the early afternoon to account for diurnal variation in blood phenylalanine levels.

## Measurement of blood analytes

The blood phenylalanine levels were measured by an enzymatic method using the Phenylalanine Assay Kit (MAK005, Millipore Sigma) according to the manufacturer's instructions. Briefly, plasma samples were deproteinized with a 10 kDa MWCO spin filter (CLS431478-25EA, Millipore Sigma) and pre-treated with 5 μL of tyrosinase for 10 min at room temperature prior to the start of the assay. Reaction mixes were made according to the manufacturer's instructions, and the fluorescence intensity of each sample was measured ($\lambda_{ex}$ = 535/$\lambda_{em}$ = 587 nm). Aspartate aminotransferase (AST) (MAK055-1KT, Millipore Sigma) and alanine aminotransferase (ALT) (MAK052-1KT, Millipore Sigma) activities were measured according to the manufacturer's instructions.

## Histology

Mice were euthanized by $CO_2$ inhalation at the time of tissue collection. Organs were harvested and fixed in 4% paraformaldehyde. After serial dehydration in ascending concentrations of ethanol and xylene, organs were paraffin-embedded and sectioned, and hematoxylin/eosin staining was performed.

## Quantitative reverse transcriptase-polymerase chain reaction (qRT-PCR)

Cells were prepared for RNA extraction with the RNeasy Mini Kit (QIAGEN) according to the manufacturer's instructions. Reverse transcription was performed using SUPERSCRIPT II (18064071, Thermo Fisher Scientific). Gene expression was measured using the following TaqMan Gene Expression Assays along with TaqMan Gene Expression Master Mix (4369016, Thermo Fisher Scientific): human B2M (Beta-2-Microglobulin) Endogenous Control (VIC/MGB probe, primer limited) (4326319E, Thermo Fisher Scientific), human PAH Exon Boundary 1-2 (Hs07288474_m1 FAM/MGB probe, Thermo Fisher Scientific), human PAH Exon Boundary 7-8 (Hs07288479_m1, FAM/MGB probe, Thermo Fisher Scientific). Each 10 μL qRT-PCR reaction contained 5 μL TaqMan Gene Expression Master Mix, 0.5 μL B2M probe, 0.5 μL probe to target the gene of interest, and 4 μL cDNA (diluted 1:10 with water) and was performed in technical duplicates. Reactions were carried out on the QuantStudio 7 Flex System (Thermo Fisher Scientific). Relative expression levels were quantified by the $2^{-\Delta\Delta Ct}$ method.

## Next-generation sequencing (NGS)

PCR reactions were performed using NEBNext Polymerase (NEB) using the primer sets listed in Supplementary Table 4, designed with Primer3 v4.1.0 (https://primer3.ut.ee/). The following program was used for all genomic DNA PCRs: 98 °C for 20 s, 35× (98 °C for 20 s, 57 °C for 30 s, 72 °C for 10 s), 72 °C for 2 min. PCR products were visualized via capillary electrophoresis (QIAxcel, QIAGEN) and then purified and normalized via an NGS Normalization 96-Well Kit (Norgen Biotek Corporation). A secondary barcoding PCR was conducted to add Illumina barcodes (Nextera XT Index Kit V2 Set A and/or Nextera XT Index Kit V2 Set D) using ≈15 ng of first-round PCR product as template, followed by purification and normalization. Final pooled libraries were quantified using a Qubit 3.0 Fluorometer (Thermo Fisher Scientific) and then after denaturation, dilution to 10 pM, and supplementation with 15% PhiX, underwent paired-end sequencing on an Illumina MiSeq System. The amplicon sequencing data were analyzed with CRISPResso2 v2[26] and scripts to quantify editing. For on-target editing, A-to-G editing was quantified at the site of the P281L variant (position 5 of the PAH1 protospacer sequence, position 4 of the PAH2 protospacer sequence) and at the site of the potential bystander adenine (position 3 of the PAH1 protospacer sequence, position 2 of the PAH2 protospacer sequence), with no other adenines present in positions 1 to 10 of either protospacer sequence. For candidate off-target sites, A-to-G editing was quantified throughout the editing window (positions 1 to 10 of the protospacer sequence). In some cases, PCR amplicons were subjected to confirmatory Sanger sequencing, performed by GENEWIZ.

## ONE-seq

ONE-seq was performed as previously described[13,21]. The human ONE-seq libraries for the PAH1 and PAH2 gRNAs were designed using the GRCh38 Ensembl v98 reference genome (ftp://ftp.ensembl.org/pub/release-98/fasta/homo_sapiens/dna/Homo_sapiens.GRCh38.dna.chromosome.{1-22,X,Y,MT}.fa, ftp://ftp.ensembl.org/pub/release-98/fasta/homo_sapiens/dna/Homo_sapiens.GRCh38.dna.nonchromosomal.fa). Sites with up to 6 mismatches and sites with up to 4 mismatches plus up to 2 DNA or RNA bulges, compared to the on-target site, were identified with Cas-Designer v1.2[27]. The final oligonucleotide sequences were generated with a script[21], and the oligonucleotide libraries were synthesized by Twist Biosciences. Recombinant ABE8.8-m protein was produced by GenScript. Duplicate ONE-seq experiments were performed with each ONE-seq library. Each library was PCR-amplified and subjected to 1.25× AMPure XP bead purification. After incubation at 25 °C for 10 min in CutSmart buffer, RNP comprising 769 nM recombinant ABE8.8-m protein and 1.54 μM gRNA was mixed with 100 ng of the purified library and incubated at 37 °C for 8 h. Proteinase K was added to quench the reaction at 37 °C for 45 min, followed by 2× AMPure XP bead purification. The reaction was then serially incubated with EndoV at 37 °C for 30 min, Klenow Fragment (New England Biolabs) at 37 °C for 30 min, and NEBNext Ultra II End Prep Enzyme Mix (New England Biolabs) at 20 °C for 30 min followed by 65 °C for 30 min, with 2× AMPure XP bead purification after each incubation. The reaction was ligated with an annealed adapter oligonucleotide duplex at 20 °C for 1 h to facilitate PCR amplification of the cleaved library products, followed by 2× AMPure XP bead purification. Size selection of the ligated reaction was performed on a BluePippin system (Sage Science) to isolate DNA of 150–200 bp on a 3% agarose gel cassette, followed by two rounds of PCR amplification to generate a barcoded library, which underwent paired-end sequencing on an Illumina MiSeq System as described above. The analysis pipeline[21] used for processing the data assigned a score quantifying the editing efficiency with respect to the on-target site to each potential off-target site. Sites were ranked based on this ONE-seq score, and the mean ONE-seq score between duplicate experiments was used for site prioritization.

## Data analysis

Sequencing data were analyzed as described above. Other data were collected and analyzed using GraphPad Prism v9.5.0.

## Reporting summary

Further information on research design is available in the Nature Portfolio Reporting Summary linked to this article.

## Data availability

DNA sequencing data that support the findings of this study have been deposited in the NCBI Sequence Read Archive under accession codes PRJNA976718 and PRJNA976729. All other data supporting the findings of this study (Figs. 1–4) are available within the Article and its Supplementary Information. The GRCh38 Ensembl v98 reference genome (ftp://ftp.ensembl.org/pub/release-98/fasta/homo_sapiens/dna/Homo_sapiens.GRCh38.dna.chromosome.{1-22,X,Y,MT}.fa, ftp://ftp.ensembl.org/pub/release-98/fasta/homo_sapiens/dna/Homo_sapiens.GRCh38.dna.nonchromosomal.fa) annotation was used. Source data are provided with this paper.

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

## Acknowledgements

This work was supported by U.S. National Institutes of Health (NIH) grants R35-HL145203 (K.M.), R01-HL148769 (K.M.), and U19-NS132301 (W.H.P., R.C.A.-N., K.M., M.-G.A., and X.W.), by an American Heart Association Career Development Award (X.W.), and by the Winkelman Family Fund in Cardiovascular Innovation (K.M.). We are grateful to Li Li and Rajan Jain for assistance with liver histology.

## Author contributions

K.M., M.-G.A., and X.W. supervised the work with intellectual input on experimental design and data analysis from W.H.P. and R.C.A.-N. D.L.B., M.C., P.Q., K.M., M.-G.A., and X.W. contributed to web laboratory experiments. K.M. performed bioinformatic analyses. M.C. and M.-G.A. contributed to mRNA production and LNP formulation. D.L.B., K.M., and X.W. drafted the manuscript, and all authors contributed to the editing of the manuscript.

## Competing interests

K.M. is an advisor to and holds equity in Verve Therapeutics and Variant Bio and is an advisor to LEXEO Therapeutics. M.-G.A. is a co-founder of and an advisor to AexeRNA Therapeutics. The University of Pennsylvania has filed a patent application related to the use of base editing for the treatment of phenylketonuria (inventors D.L.B., K.M., and X.W.). The remaining authors declare no competing interests.
