## [Peer Review File · Nature Communications]

Rapid and definitive treatment of phenylketonuria in variant-humanized mice with corrective editingREVIEWER COMMENTS

Reviewer #1 (Remarks to the Author):

Brooks et al. describe the generation and correction of a humanized mouse model for Phenylketonuria. First, they introduce the targeted mutation in a cell line with prime editing to again correct it by base editing. After selection of the base editor with the best correction rates versus bystander editing, the authors describe an in vivo correction approach using LNP. Overall, the study confirms that one of the most common human PKU-causing mutations indeed causes a phenotype in mice and that it is curable by transient base editing. While this study serves as interesting preclinical starting point for base editing for PKU, some data was quite shallow and should be repeated with more animals. The report is very concise and clear evidence is provided for the correction/base editing of a common non-cofactor responsive variant for PKU, i.e. P281L, in the mouse background harboring the humanized PAH-P281L variant. An interesting approach which is noteworthy to be shared with specialists on gene therapy, in particular interesting for metabolic liver defects in combination with LNP delivery. Nevertheless, in the light of previous demonstrations of the potential of base editing through transient LNP delivery (for PKU), there are some reservation about the originality of their approach.

General comments

It would be of interest to give a bit more detailed information on the frequency of this variant, i.e. how many PKU patients (in e.g. %) are carrying this variant and would profit from such a treatment (to become a healthy carrier for PKU if compound heterozygous or homozygous for PAH-P281L). According to the biopku database, the allele frequency of NM_000277.3: c.842C>T (p.Pro281Leu) is 2.826%. This and/or another databases should be cited.

For some treatments, only one animal was in the study (e.g. Fig 3c). I would suggest to at least repeat the experiment with one additional animal.

In order to make sure this study is reproducible; the Authors should mention the ratios at which they mixed the lipids for the LNP formulation.

Minor points

For in vitro studies with LNP, the authors state ng/mL as a dose but it would be more informative to state the dose per cell.

The authors state that previous studies needed several month until Phe levels decreased or did not achieve normalization (line 266-267). However, the authors may mention the study by Villiger et al. 2021 where curative correction of Pah was achieved in vivo using LNPs (<https://doi.org/10.1038/s41551-020-00671-z>).

Reviewer #2 (Remarks to the Author):

The authors demonstrate with outstanding clarity and conciseness the treatment of phenylketonuria (PKU) in cells and in newly developed mouse models using a therapeutic genome editing approach. This is an achievement that further opens the genome editing door to improved therapeutic options for thousands of different genetic disease. This therapy directly targets the underlying causal mutations of one of the most frequent causal mutations of this disease – and further demonstrate that (in a mouse model) the treatment of a compound heterozygote is equally effective. This study is very novel, and highly ground-breaking, and sets the foundation for a novel treatment for the subset of PKU patients with this mutation. The conclusions are justified and sound and concisely stated. The methodology is outstanding. The depth of analyses of off target edits is sufficient.

Figure 3 – Minor point: For improved clarity, the maximum value of the Y-axis could be the same for

each figure in Figure 3A, 3B, 3C, 3D. Moreover, the clarity of the findings could be improved with a log scale Y-axis.

Fig. 3e – could be improved with a log scale Y-axis to visualize the results of tissues with low levels (e.g., lung, testis)

It was initially unclear what is the second mutation was in the main text: "...compound heterozygous mice (P281L) (PKU) mice...". This is answered in the method, but for improved clarity, a very brief explanation of the second mutation would improve the main manuscript. (i.e., as simple as "... with a second mutation (delta GTAA)...")

An expanded comment on the elevated AST and ALT levels observed early in the in vivo studies would further improve the manuscript. For example, speculation on the possible cause of these transient elevations. Could they be related to the delivered mRNA, gRNA, or LNPs? Alternatively, it would be fine to NOT speculate, but then add a note that this issue remains unresolved, in the discussion after the AST and ALT results are mentioned.

Response to Reviewers' Comments

Referee #1 (Remarks to the Author):

Brooks et al. describe the generation and correction of a humanized mouse model for Phenylketonuria. First, they introduce the targeted mutation in a cell line with prime editing to again correct it by base editing. After selection of the base editor with the best correction rates versus bystander editing, the authors describe an in vivo correction approach using LNP. Overall, the study confirms that one of the most common human PKU-causing mutations indeed causes a phenotype in mice and that it is curable by transient base editing. While this study serves as interesting preclinical starting point for base editing for PKU, some data was quite shallow and should be repeated with more animals. The report is very concise and clear evidence is provided for the correction/base editing of a common non-cofactor responsive variant for PKU, i.e. P281L, in the mouse background harboring the humanized PAH-P281L variant. An interesting approach which is noteworthy to be shared with specialists on gene therapy, in particular interesting for metabolic liver defects in combination with LNP delivery. Nevertheless, in the light of previous demonstrations of the potential of base editing through transient LNP delivery (for PKU), there are some reservation about the originality of their approach.

We thank the Reviewer for the overall positive comments.

General comments

It would be of interest to give a bit more detailed information on the frequency of this variant, i.e. how many PKU patients (in e.g. %) are carrying this variant and would profit from such a treatment (to become a healthy carrier for PKU if compound heterozygous or homozygous for PAH-P281L). According to the biopku database, the allele frequency of NM_000277.3: c.842C>T (p.Pro281Leu) is 2.826%. This and/or another databases should be cited.

It is challenging to directly address this point because the allele frequency varies widely across populations, but according to the most authoritative source available on PKU (Hillert et al., 2020; ref. 2 in the manuscript), the aggregate allele frequency of *PAH* P281L is 3.2%. But this number does not account for the large portions of the world from which there are little or no data on PKU genotypes. According to the same source, the P281L allele is present in 14.8% of PKU patients in the Netherlands, 11.2% in Turkey, 10.8% in Portugal, 10.3% in Italy, and 9.4% in Germany. We have now incorporated these data in the manuscript:

However, the P281L variant, one of the most common PKU pathogenic variants with its highest prevalence in populations in the Middle East, Europe, and Russia (e.g., present in 14.8% of PKU patients in the Netherlands, 11.2% in Turkey, 10.8% in Portugal, 10.3% in Italy, and 9.4% in Germany)², is unresponsive to sapropterin (i.e., blood Phe levels do not improve with sapropterin treatment)³.

We hope this information will suffice.

For some treatments, only one animal was in the study (e.g. Fig 3c). I would suggest to at least repeat the experiment with one additional animal.

Although we had de-emphasized the work with the PAH2 guide (i.e., PAH1 is the lead drug candidate due to its more favorable off-target profile), we have now conducted an additional study with ABE8.8/PAH2 LNPs, with 2 homozygous PKU mice treated with LNPs, 3 compound heterozygous PKU mice treated with LNPs, and 3 untreated control non-PKU mice:

The results are consistent with the prior mouse data with PAH2 (we removed the original pilot mouse data previously shown for ABE8.8/PAH2 LNPs, from 1 homozygous PKU mice treated with LNPs and 1 compound heterozygous PKU mice treated with LNPs, as that was an independent experiment performed almost a year ago) as well as all of the mouse data with PAH1. We have updated the relevant text in the manuscript as follows:

We performed two additional short-term studies. In one study, we treated two 4-week-old compound heterozygous P281L (PKU) mice with ABE8.8/PAH1 LNPs, with two heterozygous P281L (non-PKU) littermates serving as controls (Fig. 3b). In the other study, we treated two 8-week-old homozygous P281L (PKU) mice and three 8-week-old compound heterozygous P281L (PKU) mice with ABE8.8/PAH2 LNPs, with three heterozygous P281L (non-PKU) littermates serving as controls (Fig. 3c). In all cases, the treated PKU mice had largely normalized blood Phe levels by 48 hours after treatment (91% and 86% mean reductions for second and third short-term studies, respectively).

In order to make sure this study is reproducible; the Authors should mention the ratios at which they mixed the lipids for the LNP formulation.

We have added this information to the Methods section:

LNPs were formulated as previously described²⁵, with the lipid components (SM-102, 1,2-distearoyl-sn-glycero-3-phosphocholine, cholesterol, and PEG-2000 at molar ratios of 50:10:38.5:1.5) being rapidly mixed with an aqueous buffer solution containing ABE8.8 mRNA and either PAH1 gRNA or PAH2 gRNA in a 1:1 ratio by weight in 25 mM sodium acetate (pH 4.0), with an N:P ratio of 5.6.

Minor points

For in vitro studies with LNP, the authors state ng/mL as a dose but it would be more informative to state the dose per cell.

We have converted the units from ng/mL to fg/cell for the in vitro studies (Fig. 1d, Fig. 1g).

The authors state that previous studies needed several month until Phe levels decreased or did not achieve normalization (line 266-267). However, the authors may mention the study by Villiger et al. 2021 where curative correction of Pah was achieved in vivo using LNPs (<https://doi.org/10.1038/s41551-020-00671-z>).

We are happy to point out the implications of this paper (now cited in the manuscript as ref. 22). We note that normalization of Phe levels was not achieved with LNPs in the paper; even after dosing and redosing with a high LNP dose (3 mg/kg), the Phe levels were greatly reduced but still did not achieve full normalization. Nonetheless, the observation that LNP redosing can increase the ultimate level of editing is an important one, as it demonstrates one key advantage of LNPs over AAV vectors. We have modified our Discussion by adding a sentence:

We note the advantages of LNP editing therapeutics over viral vector editing therapeutics, including the lack of prolonged expression of the editor—which can elicit cytolytic immune responses and exacerbate off-target editing—and lack of risk of vector sequence integration, as well as the ability to redose the therapy if needed. Indeed, a study in which *Pah^{enu2}* PKU mice were treated with LNPs to deliver a cytosine base editor into the liver did not achieve normalization of blood Phe levels with a single LNP dose, but redosing with the same LNP treatment a week later resulted in increased cumulative editing and achieved near-normalization of blood Phe levels²².

Reviewer #2 (Remarks to the Author):

The authors demonstrate with outstanding clarity and conciseness the treatment of phenylketonuria (PKU) in cells and in newly developed mouse models using a therapeutic genome editing approach. This is an achievement that further opens the genome editing door to improved therapeutic options for thousands of different genetic disease. This therapy directly targets the underlying causal mutations of one of the most frequent causal mutations of this disease – and further demonstrate that (in a mouse model) the treatment of a compound heterozygote is equally effective. This study is very novel, and highly ground-breaking, and sets the foundation for a novel treatment for the subset of PKU patients with this mutation. The conclusions are justified and sound and concisely stated. The methodology is outstanding. The depth of analyses of off target edits is sufficient.

We thank the Reviewer for the highly positive comments.

Figure 3 – Minor point: For improved clarity, the maximum value of the Y-axis could be the same for each figure in Figure 3A, 3B, 3C, 3D. Moreover, the clarity of the findings could be improved with a log scale Y-axis.

We have adjusted the Y-axis so that all are the same scale (maximum of 3200 micromol/L). We explored the use of a log scale Y-axis, but we felt that this acted to accentuate small, insignificant Phe differences at the lower end of the scale while somewhat obscuring the large and rapid Phe decreases that occurred after treatment. We show an example below (linear scale Y-axis on the left, log scale Y-axis on the right). We have stayed with linear scale Y-axes in the revised manuscript, but if the Editor and Reviewer feel strongly that log scale Y-axes are indispensable, we are happy to make that change. (Please note that all of the data displayed in the graphs are also given as raw data in a Source Data file.)

Fig. 3e – could be improved with a log scale Y-axis to visualize the results of tissues with low levels (e.g., lung, testis)

Again, we explored the use of a log scale Y-axis, but we felt that it obscured differences at the higher end of scale and actually made the data less accessible to the reader. At the same time, we appreciate the value of magnifying the data at the lower end of the range so that the reader can assess the very low levels of editing occurring in some of the non-liver tissues. Accordingly, we have used a split linear scale Y-axis, which we feel offers the best perspective for the reader. We have replaced Fig. 3e with the following, and we hope this is acceptable to the Reviewer:

It was initially unclear what is the second mutation was in the main text: “...compound heterozygous mice (P281L) (PKU) mice...”. This is answered in the method, but for improved clarity, a very brief explanation of the second mutation would improve the main manuscript. (i.e., as simple as “... with a second mutation (delta GTAA)...”

We have made the simple change suggested by the Reviewer:

In one study, we treated two 4-week-old compound heterozygous P281L (PKU) mice (with a non-humanized second mutant allele, Δ GTAA, generated by non-homologous end-joining at the P281L target site) with ABE8.8/PAH1 LNPs, with two heterozygous P281L (non-PKU) littermates serving as controls (Fig. 3b).

An expanded comment on the elevated AST and ALT levels observed early in the in vivo studies would further improve the manuscript. For example, speculation on the possible cause of these transient elevations. Could they be related to the delivered mRNA, gRNA, or LNPs? Alternatively, it would be fine to NOT speculate, but then add a note that this issue remains unresolved, in the discussion after the AST and ALT results are mentioned.

Since we are not really in a position to speculate as to the exact cause(s), we have simply added the following sentence:

AST and ALT rises are often observed with LNP treatments¹³, although the identities of the lipid and/or RNA components responsible for the rises remain unresolved.

REVIEWERS' COMMENTS

Reviewer #1 (Remarks to the Author):

The authors responded to all my comments and correspondingly answered the questions satisfactorily. I have no additional comments.

Reviewer #2 (Remarks to the Author):

My queries have been well addressed by the authors.

The authors should be commended on their thoughtful responses.

As noted in the initial review, this work is of significant importance and will make a significant impact in the field.

Recommendation: accept.